# Predictive Factors for the Successful Outcome of Urethral Sphincter Injections of Botulinum Toxin A for Non-Neurogenic Dysfunctional Voiding in Women

**DOI:** 10.3390/toxins16090386

**Published:** 2024-09-05

**Authors:** Chia-Cheng Yang, Yuan-Hong Jiang, Hann-Chorng Kuo

**Affiliations:** Department of Urology, Hualien Tzu Chi Hospital, Buddhist Tzu Chi Medical Foundation, Tzu Chi University, Hualien 970, Taiwan; ycc39946@gmail.com (C.-C.Y.); redeemer1019@yahoo.com.tw (Y.-H.J.)

**Keywords:** dysuria, urethral sphincter, voiding dysfunction, botulinum toxin A

## Abstract

Purpose: Dysfunctional voiding (DV) is not uncommon in women with non-neurogenic voiding dysfunction. Because of its unknown pathophysiology, effective and durable treatment is lacking. This study aimed to analyze the results of treatment and predictive factors for a successful outcome of botulinum toxin A (BoNT-A) treatment in female patients with DV. Methods: In total, 66 women with DV confirmed by a videourodynamic study (VUDS) were treated with a BoNT-A injection into the urethral sphincter once (n = 33) or several times (n = 33). VUDS was performed before (baseline) and after the BoNT-A treatment. Patients with a global response assessment of the voiding condition of 2 or 3 and a voiding efficiency (VE) of >20% than baseline were considered to have a successful outcome. The baseline demographics, VUDS parameters, and VUDS DV subtypes were compared between the successful and failed groups. Predictive factors for a successful outcome were investigated by logistic regression analyses. Results: Successful and failed outcomes were achieved in 27 (40.9%) and 39 (59.1%) women, respectively. After BoNT-A injections, the maximum flow rate (Qmax), voided volume, and VE all significantly increased, and the postvoid residual (PVR) was slightly improved. No significant difference in the number of injections and medical comorbidity was found between the groups. However, the successful group had a higher incidence of previous pelvic surgery. No significant difference in the treatment outcome was found among patients with different urethral obstruction sites. Significant improvements in Qmax, voided volume, PVR, VE, and the bladder outlet obstruction (BOO) index were noted in the successful group. A lower VE at baseline and a history of surgery were identified as predictive factors for a successful outcome of BoNT-A injections for treating DV. Conclusion: BoNT-A injections into the urethral sphincter can effectively improve VE in 40.9% of women with DV. Women with higher BOO grades and previous pelvic surgery are predicted to have a successful treatment outcome.

## 1. Introduction

Dysfunctional voiding (DV) in women is a condition of the lower urinary tract characterized with poor relaxation or dyssynergia of the urethral sphincter during voiding without neurological lesions or anatomical bladder outlet obstruction (BOO) [1]. The prevalence of female DV ranges from approximately 2.7% to 23% [2,3,4]. Although DV is commonly encountered in children with recurrent urinary tract infections (UTIs) and vesicoureteral reflux, the pathophysiology of female DV has not been well elucidated. Among various voiding dysfunctions in women, DV is the most challenging lower urinary tract disorder to treat [5]. Thus, effective treatment is still lacking. Patients with DV may experience difficulty urinating, large postvoid residual (PVR) urine volumes, recurrent UTIs, and bladder storage symptoms [6].

Various treatment modalities have been used to relax the urethral sphincter and pelvic floor muscles, such as training of the pelvic floor muscles, antimuscarinic therapy, sacral nerve neuromodulation, and posterior tibial nerve stimulation [4,7,8]. Given the lack of definitive medical treatment currently available for DV, clinicians have enthusiastically used urethral botulinum toxin A (BoNT-A) for this off-label indication [9]. However, a satisfactory outcome of BoNT-A has not been achievable for all patients with DV [10]. BoNT-A injections into the urethral sphincter are currently advocated for the treatment of female DV, mostly due to the simple and non-invasive procedure. Previous studies revealed that the success rate for improving voiding efficiency (VE) and the global response assessment (GRA) by one scale was approximately 70% in women with different subtypes revealed by videourodynamic study (VUDS). However, only 49.4% of women achieved a very good satisfactory outcome with a GRA of ≥2 [11,12]. Women with definite BOO might have a better successful outcome after BoNT-A injections into the urethral sphincter [13]. However, BoNT-A injections do not provide a durable treatment outcome, and patients need repeated BoNT-A injections to ensure therapeutic efficiency and maintain relief from lower urinary tract symptoms. There could be some pathophysiology underlying the factors that reduces the overall success rate of BoNT-A therapy for DV. This study retrospectively analyzed the treatment results and predictive factors for a successful outcome of BoNT-A treatment in female patients with DV.

## 2. Results

The study included a total of 66 patients. The mean age was 57.7 ± 15.6 (range, 10.4–85.1) years. Thirty-three (50%) women received only one BoNT-A injection into the urethral sphincter, and the other 33 (50%) received repeat injections. Repeat VUDS was performed 3 months after each BoNT-A injection, and the follow-up time ranged from 6 to 12 months. For patients who had several VUDS during the follow-up period, because the voiding symptoms might change over time, the best result was selected for the post-treatment evaluation, based on the lowest voiding pressure and highest maximum flow rate (Qmax). The results of the baseline and treatment VUDS are shown in Table 1. After BoNT-A injections, there were significant increases in Qmax, voided volume, and voiding efficiency (VE), with a slight improvement in postvoid residual (PVR). The baseline VUDS revealed a narrow bladder outlet at the mid-urethra in 47 women (71.2%) women, the distal urethra in 16 women (24.2%), and both the bladder neck and mid-urethra in 3 women (4.5%).

A successful outcome was achieved in 27 (40.9%) women and a failed outcome in 39 (59.1%). The follow-up periods were 7.7 ± 5.2 and 9.3 ± 6.7 months for the successful and failed groups, respectively. Table 2 shows the patients’ demographics for the successful and failed groups. No significant difference in the number of injections, medical comorbidities, and previous surgery was found between the successful and failed subgroups, except for pelvic surgery, for which the successful group had a higher incidence of previous pelvic surgery. The pelvic surgery included hysterectomy (*n* = 12), colectomy (*n* = 2), oophorectomy (*n* = 2), and cystocele repair (*n* = 2).

In the comparison of the baseline VUDS findings between the successful and failed groups, the successful group had significantly older age, lower Qmax, smaller voided volume, greater PVR, and lower VE. However, no significant differences in the treatment outcome was found among patients with BOO at the bladder neck (33.3%), mid-urethra (36.2%), or distal urethra (56.3%) (Table 3).

Table 4 shows the changes in the VUDS parameters between the failed and successful groups. As expected, the successful group had significant improvement in Qmax, voided volume, PVR, VE, and BOOI. Interestingly, only the Pdet slightly decreased, and the volume of bladder sensation did not change after BoNT-A injections. On the contrary, none of the measured parameters showed significant changes after BoNT-A injections into the urethral sphincter in the failed group.

Univariate and multivariate logistic regression analyses were conducted to identify the predictors of a successful treatment outcome, based on the baseline characteristics and VUDS parameters. A lower VE at baseline and a history of pelvic surgery were found to be predictive factors for a successful outcome of BoNT-A injections for treating DV (Table 5).

## 3. Discussion

This study revealed that a successful treatment outcome of BoNT-A injections, defined by a strict criterion of a GRA ≥2 and an improvement in VE by ≥20%, was achieved only in 40.9% of women with DV. The results were similar to our previous large-cohort retrospective analysis, i.e., a very satisfactory outcome could only be achieved in 40.9% of women with DV [11]. Moreover, a successful treatment outcome could be predicted for patients with a significantly lower baseline VE and previous pelvic surgery, suggesting the therapeutic effect of BoNT-A on dysfunction of the urethral sphincter is for true external sphincter spasticity but not other causes of voiding dysfunction in women.

The pathophysiology of non-neurogenic DV in women has not been well elucidated. Pelvic floor muscle overactivity or hypertonicity may be caused by inflammatory conditions of the pelvic organ, denervation and re-innervation of the pelvic floor, or a learned behavior [14,15,16,17]. In severe female DV, severe storage symptoms, recurrent urinary tract infections, chronic urine retention, or upper urinary tract deterioration may ensue [18]. Poor relaxation of the pelvic floor muscles is also commonly encountered in women with DV [14,15]. Dysfunctional pubococcygeal muscles could also contribute to the development of DV [17]. Differential diagnosis between true dysfunction of the urethral sphincter and dysfunction of the pelvic floor muscles is not easy, even by VUDS [12]. Thus, effective and durable treatment of DV has not currently been established. In addition to medical therapy, biofeedback-based pelvic floor muscle exercise, posterior tibial nerve stimulation, and BoNT-A injections into the urethral sphincter have proven effective in some women with DV; however, these are not durable treatment options [4,7,8,9]. Thus, repeated BoNT-A treatment is needed to ensure longer efficacy.

BoNT-A injections into the urethral sphincter have not been licensed for the treatment of voiding dysfunction caused by a dyssynergic urethral sphincter or DV; however, enthusiasm has been aroused because early clinical trials have demonstrated its efficacy and safety [9,10,11,12]. The rates of successful treatment outcomes vary widely, from 40% to 70%, depending on the definition of DV in the patients’ selection criteria and the determination of a successful treatment outcome [10,11,12]. Patients with DV may present with a narrow urethral lumen at the middle or distal portion during voiding. Some patients may present with a tight bladder neck initially and a tight mid-urethra after TUI-BN. These different clinical phenotypes might have different pathophysiologies; therefore, the treatment outcome might vary mildly. Patients who have both BND and DV were the best treated among three DV subgroups [11]. Regarding the pathophysiology of non-neurogenic voiding dysfunction, the dyssynergic urethral sphincter is not likely caused by suprasacral dysregulation between the detrusor and pudendal nucleus, such as in patients with spinal cord injury or multiple sclerosis. The DV might be a result of neurological dysregulation at the sacral cords or pelvic ganglionic level, which would also explain the effectiveness of sacral neuromodulation in modulating the micturition reflex circuit in the CNS and restoring the coordination between the detrusor and urethral sphincter during micturition [19,20].

BoNT-A injections into the urethral sphincter can result in the relaxation of the striated muscle by inhibiting the release of acetylcholine from nerve terminals and might inhibit the release of neuropeptides in the dorsal horn nucleus because of chronic inflammation [21]. With this dual mechanism of action, BoNT-A injections could result in relaxation of the urethral sphincter during voiding and increase the Qmax and VE [10]. A single BoNT-A injection might not resolve the chronic inflammation in the dorsal horn ganglia; therefore, repeated BoNT-A injections into the urethral sphincter might have a chance of long-term efficacy [22,23]. Although the theoretical benefits of BoNT-A injections for the urethral sphincter seem reasonable, the present and previous studies have revealed that this treatment goal is still not achievable in over half of the patients with DV. These patients often still require clean intermittent catheterization to evacuate a large PVR and prevent UTIs [10]. The pathophysiology of the successful and failed groups might be different, which needs further investigation.

Dysregulated urethral function with spastic or a non-relaxing external urethral sphincter has been considered a cause of DV, leading to voiding symptoms, such as slow urinary flow and a large PVR [15,24]. However, the actual mechanism for DV is not well understood. It is possible that different pathophysiologies might exist in hypertonicity of the female urethral sphincter or pelvic floor, which might result in DV with varying degrees of urethral narrowing during voiding. Therefore, the reduction in the urethral sphincter’s hypertonicity by BoNT-A injections and the resumption of spontaneous voiding usually cannot achieve a highly satisfactory rate through a single injection. Many women might have a mild improvement (GRA = 1) and a small increase in VE (10%), and only 40% of the patients achieved good satisfaction. The suboptimal treatment outcome of female DV is also possibly caused by chronic inflammation in the CNS and sensitization, leading to the persistent sensitization and hyperactivity of the urethral sphincter. Under this consideration, we hypothesize that repeat BoNT-A injections into the urethral sphincter might help adequately alleviate the inflammation of the CNS through the sensory nerves of the pudenda, leading to a long-term relaxation of the urethral sphincter. However, in this study, no significant difference was found in the number of BoNT-A injections regarding a successful outcome. Therefore, the exact mechanism for effective BoNT-A injections for treating DV remains unknown.

This study revealed that the successful group had a higher BOO grade, including higher Pdet, lower Qmax, larger PVR, and lower VE during VUDS. This finding was compatible with the results of our previous cohort studies, reporting that a higher BOO grade indicated a successful BoNT-A treatment outcome [11,12,13]. Interestingly, this group also had a higher incidence of previous pelvic surgery. The reason for this association between pelvic surgery and a higher BOO grade is unknown; however, the results imply that a dyssynergic urethral sphincter may be directly or indirectly related to the dysregulation of the sympathetic and parasympathetic nerves in the pelvic nerve plexus after previous pelvic organ surgery. Previous studies have shown that a crosstalk between the pelvic organs is present after long-term sensitization of the pelvic organ between the lower urinary tract and the colon [25]. The inflammation-induced sensory afferent activation from the pelvic organ’s wound may contribute to cross-sensitization, such as in patients with bladder pain syndrome or hypertonicity of the pelvic floor muscles [26]. Spinal cord inflammation may also develop after pelvic organ surgery; therefore, patients might have DV and a higher BOO grade during voiding.

Contrastingly, the pathophysiology of DV has not been fully elucidated. Several theories have accounted for DV in women and children. In addition to the association with encopresis in children, dysregulation of the CNS or learned habits have been considered to cause DV in children and women, which could be effectively treated by biofeedback therapy [27,28]. Because the causes of DV might not only originate from the urethral sphincter’s hypertonicity but also from dysfunctions of the pelvic floor muscles, BoNT-A injections into the urethral sphincter could not effectively decrease the muscular tone and adequately relieve the bladder outlet’s resistance. Therefore, the BOOI in the failed group was relatively lower than that in the successful group, and changes in the BOOI were also not significant after BoNT-A injections into the urethral sphincter in the failed group. The results of this study further emphasize that BoNT-A injections into the urethral sphincter are effective in reducing urethral resistance only in women with evident BOO because of a hypertonic or dyssynergic urethral sphincter [13,29]. Patients who are not likely to have BOO caused by a dysfunctional urethral sphincter, as manifested by a lower Pdet, higher Qmax, and a higher VE, might not benefit from BoNT-A injections into the urethral sphincter.

The limitations of this study include the small number of cases, the retrospective analysis, and the non-randomized design. Additionally, the follow-up period extending beyond 6 months after the latest BoNT-A injection suggested that the effects of BoNT-A injections into the urethral sphincter may not be durable. The possible reason is that the underlying pathophysiology of DV is not adequately resolved by BoNT-A injections into the urethral sphincter. Given the unsatisfactory success rate, the purpose of this study aimed to investigate any predictive factors for selecting appropriate patients for BoNT-A injections in the future. Further investigations into the electrophysiology and autonomic dysregulation of DV might help urologists understand this disease and provide more effective treatment.

## 4. Conclusions

This study revealed that BoNT-A injections into the urethral sphincter can effectively improve VE and GRA in only 40.9% of women with DV. Women with a successful treatment outcome had higher BOO grades, lower VE, and previous pelvic surgery, whereas patients with treatment failure had lower BOO grades and higher VE.

## 5. Methods

From September 2020 to September 2023, 66 consecutive women with voiding dysfunction received BoNT-A injections into the urethral sphincter for a clinical and VUDS-confirmed diagnosis of DV. This retrospective study analyzed the treatment outcomes. All patients had undergone a VUDS examination and were found to have high voiding pressure, a low maximum flow rate (Qmax), and a narrow bladder neck, mid-urethra, or distal urethra, as demonstrated by voiding cystourethrography [12]. The participants who had previously been treated with an alpha-blocker and baclofen without success then received BoNT-A injections into the urethral sphincter. Patients who had urodynamic detrusor overactivity (DO) were also treated with antimuscarinics or beta-3 adrenoceptor agonists. Patients with voiding dysfunction due to SCI, a cerebral vascular accident (CVA), multiple sclerosis, or peripheral neuropathy were classified as having neurogenic voiding dysfunctions and were excluded from this study.

This study was approved by the institutional review board of the authors’ hospital (IRB: 110-265-A, dated 1 July 2022). Informed consent was waived due to the retrospective study design. All study procedures were in accordance with the Declaration of Helsinki.

The videourodynamic parameters of the VUDS, including the status of the bladder neck during voiding cystourethrography, first bladder sensation (FSF), cystometric bladder capacity (CBC), detrusor pressure (Pdet), maximum flow rate (Qmax), PVR, and abdominal pressure to void, were recorded and analyzed [6]. The terminologies used in this study were in accordance with the recommendations of the International Continence Society [30]. DO that occurred during the bladder storage phase and before uninhibited voiding were recorded as phasic DO and terminal DO, respectively. The VE (defined as voided volume/CBC), the BOO index (BOOI, defined as Pdet–2 × Qmax), and the bladder contractility index (BCI, defined as Pdet + 5 × Qmax) were calculated from the measured parameters. During voiding, the appearance of the bladder outlet was carefully evaluated. DV was defined by a high voiding Pdet (≥35 cmH_2_O), a low Qmax < 15 mL/s, and urethral narrowing at the mid-urethra (Figure 1A) or distal urethra (Figure 1B) [1,31]. Patients with a history of transurethral incision for bladder neck dysfunction (BND) (Figure 1C) but who were found to have mid-urethral BOO during a follow-up VUDS were specifically categorized as having BND plus DV. (Figure 1D) Electromyography (EMG) of the urethral sphincter was performed using surface EMG patches to record the pelvic floor muscles’ activity during voiding, which could show intermittency, hyperactivity, or non-relaxation [29]. Repeat VUDS was performed at 6–12 months after the initial diagnosis of DV.

The procedure of BoNT-A injections into the urethral sphincter followed the protocol outlined in our previous reports [10]. In total, 100 U of onabotulinum toxin A was dissolved into 5 mL of a normal saline solution. Under intravenous general anesthesia, the BoNT-A solution was injected into the urethral sphincter along the urethral lumen at the 2, 5, 7, 10, and 12 o’clock positions by a single author (HCK). The injecting needle was inserted into the urethral sphincter at each injection site to depths of 2.0 cm and 1.0 cm, with 0.5 mL of the BoNT-A solution injected into each site. Following the BoNT-A injections, the urine was drained, and the patients were discharged after they had awakened and were free of adverse events. Patients were then followed up at the outpatient clinic to assess changes in voiding symptoms, Qmax, and PVR. Medical therapy with alpha-blockers with or without baclofen was continued as needed.

When patients still experienced voiding difficulty, low Qmax, and large PVR after the initial BoNT-A injection, repeated BoNT-A injections into the urethral sphincter were recommended at least 6 months after the prior injection Some patients preferred to stay on medical therapy and declined repeat BoNT-A injections. VUDS was repeated to compare the storage and voiding parameters and assess the treatment results of the BoNT-A injections. The treatment outcome was assessed 6 months after the last urethral BoNT-A injection. Improvement in voiding symptoms was evaluated by the GRA, defined as markedly worse (−3), moderately worse (−2), mildly worse (−1), no change (0), mildly improved (+1), moderately improved (+2), and markedly improved (+3) [32]. Patients with a GRA of two or three points and a post-treatment improvement in VE of 20% or more compared with the baseline were considered to have a successful outcome. Otherwise, the treatment was considered a failure. Patients who did not respond to urethral BoNT-A injections were recommended to undergo additional therapy, including sacral neuromodulation (*n* = 2), pelvic floor muscle biofeedback therapy (n = 5), or transcutaneous tibial nerve stimulation (n = 3). However, these treatments did not yield satisfactory results for all patients. Continuous variables are expressed as means with standard deviations, and categorical data are presented as numbers and percentages. The chi-square test for categorical variables and the Wilcoxon rank-sum test for continuous variables were used to determine *p*-values between groups for statistical comparisons. All assessments were two-sided and considered significant at *p* < 0.05. All calculations were performed using SPSS for Windows, version 16.0 (SPSS, Chicago, IL, USA).

## Figures and Tables

**Figure 1 toxins-16-00386-f001:**
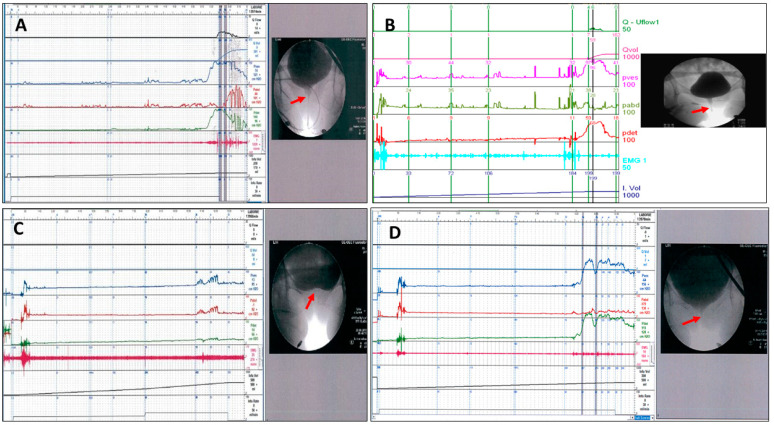
VUDS characteristics of bladder outlet obstruction in women with dysfunctional voiding (DV), with urethral narrowing (arrow) at the (**A**) mid-urethra, (**B**) distal urethra, (**C**) bladder neck before transurethral incision of the bladder neck (TUI-BN), and (**D**) DV at the mid-urethra after TUI-BN of the woman shown in Figure 1C.

**Table 1 toxins-16-00386-t001:** Baseline and post-treatment (BoNT-A) urodynamic findings.

VUDS Parameter	Baseline	Post-BoNT-A	*p*-Value
FSF (mL)	124.1 ± 66.6	132.9 ± 59.9	0.389
FS (mL)	200.8 ± 90.2	212.9 ± 95.8	0.334
US (mL)	241.2 ± 106.1	241.2 ± 105.8	0.998
Pdet (cmH_2_O)	49.4 ± 27.8	44.3 ± 30.9	0.167
Qmax (mL/s)	9.2 ± 6.9	12.3 ± 7.8	<0.001
Volume (mL)	144.8 ± 106.9	209.0 ± 121.0	<0.001
PVR (ml)	132.9 ± 155.6	100.9 ± 109.3	0.052
VE	0.58 ± 0.30	0.69 ± 0.27	0.002
BOOI	32.5 ± 31.9	26.6 ± 33.7	0.104
BCI	91.7 ± 36.2	88.7 ± 43.0	0.605

BoNT-A, botulinum toxin A; VUDS, videourodynamic study; FSF, first sensation of filling; FS, full sensation; US, urgency sensation; Pdet, detrusor pressure; Qmax, maximum flow rate; PVR, postvoid residual; VE, voiding efficiency; BOOI, bladder outlet obstruction index; BCI, bladder contractility index.

**Table 2 toxins-16-00386-t002:** Baseline demographics between successful and failed treatment outcomes.

	Failure (*n* = 39)	Successful (*n* = 27)	*p*-Value
**BoNT-A injection(s)**			
Single injection	18 (46.2%)	15 (55.6%)	0.453
Repeat injections	21 (53.8%)	12 (44.4%)	
**Comorbidities**			
Hypertension	13 (33.3%)	14 (51.9)	0.132
Diabetes mellitus	8 (20.5%)	7 (25.9%)	0.606
Coronary artery disease	4 (10.3%)	2 (7.4%)	1.000
Chronic kidney disease	2 (5.1%)	2 (7.4%)	1.000
Cerebrovascular disease	1 (2.6%)	1 (3.7%)	1.000
Dementia	2 (5.1%)	0	0.509
Congestive heart failure	2 (5.1%)	0	0.509
**Previous surgeries**			
TUI-BN	9 (23.1%)	3 (11.1%)	0.332
Spinal surgery	4 (10.3%)	3 (11.1%)	1.000
Pelvic surgery	6 (15.4%)	12 (44.4%)	0.009

BoNT-A, botulinum toxin A; TUI-BN, transurethral incision of the bladder neck.

**Table 3 toxins-16-00386-t003:** Comparison of VUDS findings between failure and successful treatment outcomes.

VUDS Parameter	Failure (*n* = 39)	Successful (*n* = 27)	*p*-Value
Follow-up period	9.3 ± 6.7	7.7 ± 5.2	0.295
Age (years)	54.4 ± 16.4	62.4 ± 13.1	0.038
FSF (mL)	114.3 ± 52.7	138.2 ± 81.7	0.154
FS (mL)	192.8 ± 84.3	212.4 ± 98.6	0.388
US (mL)	230.1 ± 97.1	257.3 ± 117.8	0.310
Pdet (cm H_2_O)	45.4 ± 27.2	55.2 ± 28.2	0.164
Qmax (mL/s)	10.7 ± 7.3	7.1 ± 6.0	0.038
Volume (mL)	177.8 ± 116.4	97.1 ± 69.0	0.002
PVR (mL)	70.4 ± 99.7	223.2 ± 177.9	<0.001
VE	0.74 ± 0.23	0.35 ± 0.25	<0.001
BOOI	27.8 ± 32.0	39.3 ± 31.2	0.151
BCI	89.6 ± 35.6	94.8 ± 37.4	0.569
VUDS DV subtype			
Mid-urethral	30 (76.9%)	17 (63.0%)	0.353
Distal urethral	7 (17.9%)	9 (33.3%)	
BND	2 (5.1%)	1 (3.7%)	

VUDS, videourodynamic study; FSF, first sensation of filling; FS, full sensation; US, urgency sensation; Pdet, detrusor pressure; Qmax, maximum flow rate; PVR, postvoid residual, VE, voiding efficiency; BOOI, bladder outlet obstruction index; BCI, bladder contractility index; DV, dysfunctional voiding; BND, bladder neck dysfunction.

**Table 4 toxins-16-00386-t004:** Changes in the VUDS parameters between patients with failed or successful treatment outcomes.

VUDS Parameter		Failure (*n* = 39)	*p*-Value	Successful (*n* = 27)	*p*-Value	Δ*p*-Value
FSF (mL)	BL F/U	114.3 ± 52.7 128.9 ± 53.7	0.183	138.2 ± 81.7 138.7 ± 68.6	0.976	0.505
FS (mL)	BL F/U	192.8 ± 84.3 202.4 ± 86.2	0.495	212.4 ± 98.6 228.0 ± 108.0	0.503	0.818
US (mL)	BL F/U	230.1 ± 97.1 233.3 ± 104.0	0.813	257.3 ± 117.8 252.5 ± 109.3	0.842	0.771
Pdet (cm H_2_O)	BL F/U	45.4 ± 27.2 42.9 ± 31.2	0.633	55.2 ± 28.2 46.3 ± 31.0	0.079	0.394
Qmax (mL/s)	BL F/U	10.7 ± 7.3 10.9 ± 7.9	0.777	7.1 ± 6.0 14.4 ± 7.4	<0.001	<0.001
Volume (mL)	BL F/U	177.8 ± 116.4 200.0 ± 130.7	0.206	97.1 ± 69.0 222.1 ± 106.4	<0.001	0.001
PVR (mL)	BL F/U	70.4 ± 99.7 98.1 ± 112.3	0.061	223.2 ± 117.9 104.9 ± 106.8	<0.001	<0.001
VE	BL F/U	0.74 ± 0.23 0.68 ± 0.28	0.016	0.35 ± 0.25 0.70 ± 0.24	<0.001	<0.001
BOOI	BL F/U	27.8 ± 32.0 27.7 ± 31.5	0.990	39.3 ± 31.2 24.9 ± 37.3	0.002	0.050
BCI	BL F/U	89.6 ± 35.6 81.0 ± 46.3	0.265	94.8 ± 37.4 99.8 ± 35.6	0.571	0.248

VUDS, videourodynamic study; BL, baseline; F/U, follow-up; FSF, first sensation of filling; FS, full sensation; US, urgency sensation; Pdet, detrusor pressure; Qmax, maximum flow rate; PVR+ postvoid residual; VE, voiding efficiency; BOOI, bladder outlet obstruction index; BCI, bladder contractility index.

**Table 5 toxins-16-00386-t005:** Univariate and multivariate logistic regression analyses of the predictors of a successful treatment outcome for the baseline characteristics and VUDS parameters.

	Univariate		Multivariate	
Variables	OR (95% CI)	*p*-Value	OR (95% CI)	*p*-Value
Age (years)	1.037 (1.001–1.075)	0.043	0.990 (0.945–1.038)	0.687
FSF (mL)	1.006 (0.998–1.014)	0.169		
FS (mL)	1.002 (0.997–1.008)	0.384		
US (mL)	1.002 (0.998–1.007)	0.306		
Pdet (cm H_2_O)	1.013 (0.995–1.032)	0.165		
Qmax (mL/s)	0.913 (0.834–0.999)	0.048	1.047 (0.920–1.191)	0.486
VE	0.003 (0.000–0.045)	<0.001	0.001 (0.000–0.034)	<0.001
BOOI	1.012 (0.996–1.028)	0.153		
BCI	1.004 (0.990–1.018)	0.563		
VUDS DV subtype		0.365		
Mid-urethral	Ref.			
Distal urethral	2.269 (0.716–7.188)			
BND	0.882 (0.074–10.464)			
**Botox injection(s)**		0.453		
Single injection	Ref.			
Repeat injections	0.686 (0.256–1.838)			
**Comorbidities**				
Hypertension	2.154 (0.787–5.893)	0.135		
Diabetes mellitus	1.356 (0.425–4.325)	0.607		
CAD	0.700 (0.119–4.123)	0.693		
CKD	1.480 (0.195–11.208)	0.704		
CVA	1.462 (0.087–24.430)	0.792		
Dementia	0.000 (0.000)	0.999		
CHF	0.000 (0.000)	0.999		
**Previous surgeries**				
TUI-BN	0.417 (0.101–1.711)	0.224		
Spinal surgery	1.094 (0.224–5.334)	0.912		
Pelvic surgery	4.400 (1.387–13.959)	0.012	6.643 (1.223–36.073)	0.028

VUDS, videourodynamic study; FSF, first sensation of filling; FS, full sensation; US, urgency sensation; Pdet, detrusor pressure; Qmax, maximum flow rate; PVR, postvoid residual; VE, voiding efficiency; BOOI, bladder outlet obstruction index; BCI, bladder contractility index; BND, bladder neck dysfunction; CAD, coronary artery disease; CKD, chronic kidney disease; CVA, cerebrovascular disease; CHF, congestive heart failure; DVL, dysfunctional voiding; BND, bladder neck dysfunction; TUI-BN, transurethral incision of the bladder neck.

## Data Availability

Data of this study are available by contacting the corresponding author.

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
