# Peer review of "Predictive Factors for the Successful Outcome of Urethral Sphincter Injections of Botulinum Toxin A for Non-Neurogenic Dysfunctional Voiding in Women"

_toxins, 2024, doi:10.3390/toxins16090386_

Round 1

Reviewer 1 Report

Comments and Suggestions for Authors

The authors describe in a retrospective study the resulta of BTX injections in the urethral sphincter in women with DV.

Although interesting, I have some concerns, mainly about the correct diagnosis of DV, the definition of success and the interpretation of the results.

Line 38 the definition of DV is non neurogenic, so geriatric patients with CNS lesions should not be diagnosed as having DV

Lin 67: only midurethral narrowing corresponds to a non relaxing (external) sphincter

Figure 1

B and C do not show dyssynergia on EMG. How solid is the diagnosis of DV?

Lin e115: VE improvement of 20% is a very low threshold for success.

Apparently, the injections wer done under general anesthesia, which is not explicitly mentioned.

Voided volume increases from 145 to 209 ml. According to the Liverpool nomograms, with increasing voided volume, Qmax will increase accordingly, also  with stable outflow resistance. So the question is whether there is in fact a decrease in BOOI, or only the effect of the increase in voided volume.

May there be an effect of BTX on the detrusor muscle?

Table 3

Only midurethral obstruction is supposed to be due a non relaxing sphincter. In this group, 17/30 treatments were successful. Patients with non relaxing bladder neck or distal obstruction probably have another diagnosis

Line 246 I am not familiar with the relationship between chronic inflammation of the CNS and DV. In my opinion will a neurogenic cause  exclude the diagnosis of idiopathic DV

Author Response

The authors describe in a retrospective study the results of BTX injections in the urethral sphincter in women with DV. Although interesting, I have some concerns, mainly about the correct diagnosis of DV, the definition of success and the interpretation of the results.

Line 38 the definition of DV is non neurogenic, so geriatric patients with CNS lesions should not be diagnosed as having DV

Reply: Thank you for the comment. Based on the definition of DV, we have deleted the statement that “geriatric patients with central nervous system (CNS) lesions and voiding dysfunction” in line 38, accordingly.

Lin 67: only midurethral narrowing corresponds to a non relaxing (external) sphincter

Reply: Thank you for the comment. The findings of videourodynamic study of female DV revealed the urethral narrowing located not only at the mid-urethral narrowing. Some women may have distal urethral narrowing or have an initial bladder neck narrowing and a mid-urethral narrowing after TUI-BN. (as shown in the revised figure 1C and 1D) According to the IUGA/ICS joint report, dysfunctional voiding is characterized by an intermittent and/or fluctuating flow rate due to involuntary intermittent contractions of the peri-urethral striated (mid-urethra) or levator muscles (distal urethra) during voiding in neurologically normal women. (Neurourol Urodyn. 2010;29(1):4-20.) Therefore, women with DV may have bladder outlet narrowing at different site, with a high voiding pressure. We have stated this VUDS finding in the Methods section, with references. (Lines 88-93)

Figure 1, B and C do not show dyssynergia on EMG. How solid is the diagnosis of DV?

Reply: Thank you for the comment. The EMG study in VUDS is usually performed by surface EMG, which might not exactly reflect the true urethral sphincter hyperactivity during voiding. EMG alone would have given the wrong diagnosis in 20.6% of those with DV (false negative) and when fluoroscopy is used to define these entities, the accuracy of EMG to differentiate them is questionable. (ref. 17. Urology. 2012;80:55-60) Therefore, in the diagnosis of female DV, a sustained detrusor contractility with a low Qmax and narrow urethra are necessary to achieve a solid diagnosis. (ref. 15. J Urol 2001; 165: 143-7) Pelvic floor EMG could be intermittent, hyperactivity, or non-relaxation, (ref. 16. Neurourol Urodyn. 2010;29(1):4-20.) as shown in the figure 1. (Lines 93-96) In the original figure 1C, we showed a woman with tight BN and detrusor underactivity before TUI-BN. In the revised manuscript, we have added a figure 1D to show the VUDS of this woman with DV after TUI-BN. (Lines 99-102)

Lin e115: VE improvement of 20% is a very low threshold for success.

Reply: Thank you for the comment. We agree that the current treatment results of Botox urethral sphincter injections on female DV was not satisfactory. Even with a satisfactory result defined by VE improvement of 20%, urethral sphincter BoNT-A injections can only effectively improve VE in 40.9% of women with DV. (Lines 307-308)

Apparently, the injections were done under general anesthesia, which is not explicitly mentioned.

Reply: Thank you for the comment. We have added the statement that urethral BoNT-A was injected under intravenous general anesthesia. (Line 105)

Voided volume increases from 145 to 209 ml. According to the Liverpool nomograms, with increasing voided volume, Qmax will increase accordingly, also with stable outflow resistance. So the question is whether there is in fact a decrease in BOOI, or only the effect of the increase in voided volume.

Reply: Thank you for the comment. In Table 1, the Qmax increased from 9.2 ml/s to 12.3 ml/s, we also showed that voided volume increased from 145 to 209ml, and PVR decreased from 133 to 101ml. If we calculated the corrected Qmax, the cQmax also increased from 0.55 to 0.703, indicating that the uroflow did increase after Botox treatment. The improvement in Qmax or cQmax might in part from decrease in BOOI (though not significant) and part from increased voided volume because of reduced BOO resistance and decreases bladder overactivity (significant).

May there be an effect of BTX on the detrusor muscle?

Reply: Thank you for the comment. I have no idea on this point of view. However, if urethral Botox injection had effect on detrusor muscle, the PVR should increase, which was not observed in the results of this study.

Table 3, Only midurethral obstruction is supposed to be due a non relaxing sphincter. In this group, 17/30 treatments were successful. Patients with non relaxing bladder neck or distal obstruction probably have another diagnosis

Reply: Thank you for the comment. The findings of videourodynamic study of female DV revealed the urethral narrowing located not only at the mid-urethra but also at the distal urethra. Some women may have distal urethral narrowing or have an initial bladder neck narrowing and a mid-urethral narrowing after TUI-BN. According to the IUGA/ICS joint report, dysfunctional voiding is characterized by an intermittent and/or fluctuating flow rate due to involuntary intermittent contractions of the peri-urethral striated or levator muscles during voiding in neurologically normal women. (Neurourol Urodyn. 2010;29(1):4-20.) We agree that DV with urethral narrowing at different site might have different pathophysiology, in this study, no significant difference in the successful treatment outcome was found among patients with BOO at the bladder neck (33.3%), mid-urethra (36.2%), or distal urethra (56.3%) We will discuss this point in the Discussion section. It is possible that different pathophysiology might exist in female urethral sphincter or pelvic floor hypertonicity, which might result in DV with different urethral narrowing during voiding. (Lines 252-254)

Line 246 I am not familiar with the relationship between chronic inflammation of the CNS and DV. In my opinion will a neurogenic cause exclude the diagnosis of idiopathic DV

Reply: Thank you for the comment. The statement of DV and CNS inflammation and sensitization is a hypothetical point-of view. BoNT-A injection had been applied in treatment of neuropathic pain and neurogenic inflammation. (Schmerz. 2003 Apr;17(2):149-65.; Plast Reconstr Surg Glob Open. 2018 Oct 16;6(10):e1847.) Based on these clinical evidence, we hypothesized that the hyperactive urethral sphincter in DV might be caused by central inflammation and sensitization. Therefore, repeated BoNT-A injections are necessary to desensitize the inflammation and resume normal urethral sphincter activity. (Lines 258-260)

Reviewer 2 Report

Comments and Suggestions for Authors

The objective of the study is to identify predictive factors for the success of botulinum toxin A injection in women with dysfunctional voiding (DV). Numerous studies over the years have identified urethral botulinum toxin injection as a possible off-label treatment.

Review the grammar of the title.

Introduction: The diagnostic process that led to the identification of DV in women examined is not explained. In the definition of the study's success, reference is made to improvement in voiding efficiency (VE) and Global Response Assessment (GRA), as well as videourodynamic subtypes that are not explained. The duration of the outcomes resulting from botulinum toxin injection is discussed, but not explained.

Methods: Was the physician who performed the infiltrations always the same?

Were all participants treated with α-blockers and baclofen?

What evaluation nomogram for obstruction was used in VUDS?

Better explanation of the BoNT-A injection technique is needed.

How many women had undergone additional therapies for the treatment of DV, and what were those therapies?

Results: The study includes a total of 66 patients, aged 57.7 +/- 15.6 years (range, 10.4-85.1), with an age range that is too wide to speak of a homogeneous population.

Reference is made to the role of chronic inflammation in the onset of DV. What is the minimum number of injections after which an improvement is perceived?

Study Weaknesses: It is a retrospective study; the study population is small; the follow-up is short.

Conclusions: BoNT-A injection at the urethral sphincter improves VE in 40.9% of women with DV, with greater improvement for higher-grade BOO and previous pelvic surgery. Were differences recognized between the three BOO groups considered? For example, literature studies recognize a higher success rate for BND plus DV compared to others.

In fact, can we claim that the success rate of botulinum toxin A administration at the urethral level is statistically higher than non-administration? In the literature, for example, there are articles that argue there is no statistically significant difference between botulinum toxin administration and placebo, or that medical treatment with or without urethral botulinum toxin injection resulted in a reduction in Pdet and BOOI.

In fact, the conclusions reached by the study are redundant with what is already present in the literature, except for the link between previous pelvic surgery and results of botulinum toxin A administration.

Comments on the Quality of English Language

Review the grammar of the title.

Author Response

The objective of the study is to identify predictive factors for the success of botulinum toxin A injection in women with dysfunctional voiding (DV). Numerous studies over the years have identified urethral botulinum toxin injection as a possible off-label treatment.

Review the grammar of the title.

Reply: Thank you for the comment. We have revised the title “ ….. in women”. (Line 4)

Introduction: The diagnostic process that led to the identification of DV in women examined is not explained. In the definition of the study's success, reference is made to improvement in voiding efficiency (VE) and Global Response Assessment (GRA), as well as videourodynamic subtypes that are not explained. The duration of the outcomes resulting from botulinum toxin injection is discussed, but not explained.

Reply: Thank you for the comment. We have stated the definition of DV in women, in the beginning of Introduction, (Lines 33-35) and added the definition of DV based on the VUDS findings. (Lines 88-96) In the original figure 1C, we showed a woman with tight BN and detrusor underactivity before TUI-BN. In the revised manuscript, we have added a figure 1D to show the VUDS of this woman with DV after TUI-BN. The treatment outcome was assessed at 6 months after the last urethral BoNT-A injection. (Lines 119) GRA assessment was reported in the Method section. (Lines 120-122)

Methods: Was the physician who performed the infiltrations always the same?

Reply: Thank you for the comment. The urethral sphincter BoNT-A injection was performed solely by the single author (HCK). (Lines 107)

Were all participants treated with α-blockers and baclofen?

Reply: Thank you for the comment. In the treatment protocol, all participants who had been treated with alpha-blocker and baclofen but failed then received urethral sphincter BoNT-A injections. (Lines 68-70)

What evaluation nomogram for obstruction was used in VUDS?

Reply: Thank you for the comment. We defined female dysfunctional voiding by a high voiding Pdet ≥ 35 cmH2O, low Qmax <15 ml/s, and urethral narrowing at the mid-urethra or distal urethra. (Lines 88-90) There have been different definitions for female BOO and DV. We have used this definition for >10 years in clinical diagnosis and management of women with DV. (Ref. 10-12)

Better explanation of the BoNT-A injection technique is needed.

Reply: Thank you for the comment. The urethral BoNT-A injection technique has been stated in the Methods section. (Lines 104-109)

How many women had undergone additional therapies for the treatment of DV, and what were those therapies?

Reply: Thank you for the comment. Patients who failed urethral BoNT-A injection were recommended to undergo additional therapy, including sacral neuromodulation (n=2), pelvic floor muscle biofeedback therapy (n=5), or transcutaneous tibial nerve stimulation (n= 3). However, the treatment outcome was not satisfactory in all patients. (Lines 125-128)

Results: The study includes a total of 66 patients, aged 57.7 +/- 15.6 years (range, 10.4-85.1), with an age range that is too wide to speak of a homogeneous population.

Reply: Thank you for the comment. This was a retrospective analysis of treatment outcome of urethral BoNT-A injection for female DV. The patients were consecutively enrolled, therefore, the age distribution was not homogeneous.

Reference is made to the role of chronic inflammation in the onset of DV. What is the minimum number of injections after which an improvement is perceived?

Reply: Thank you for the comment. As shown in Table 2, 15/33 (55.6%) patients had a successful result after single BoNT-A injection, whereas only 12/33 (44.4%) patients had a successful result after repeat injections. This result indicates that the pathophysiology for DV in women might be different. (Lines 252-254) Although chronic inflammation in DV is speculated, we do not have evidence to support this hypothesis. (Lines 258-260)

Study Weaknesses: It is a retrospective study; the study population is small; the follow-up is short.

Reply: Thank you for the comment. We agree that the study was a retrospective analysis, with small case number and short follow-up period. However, female DV is not very common in women with voiding dysfunction and patients might not accept urethral BoNT-A injection as their long-term treatment. Because the success rate is not satisfactory, the purpose of this study aims to investigate any predictive factors for selecting appropriate patients for this treatment in the future. We have added this point in the limitation of the study. (Lines 301-303)

Conclusions: BoNT-A injection at the urethral sphincter improves VE in 40.9% of women with DV, with greater improvement for higher-grade BOO and previous pelvic surgery. Were differences recognized between the three BOO groups considered? For example, literature studies recognize a higher success rate for BND plus DV compared to others.

Reply: Thank you for the comment. In this study, we found no significant difference in the successful treatment outcome was found among patients with BOO at the bladder neck (33.3%), mid-urethra (36.2%), or distal urethra (56.3%) (Lines 166-168). Because the patient number is small in this study, the results might not reflect the true scenario of BoNT-A treatment for female DV. The possible reason is that the underlying pathophysiology of DV is not adequately resolved by urethral sphincter BoNT-A injections.

In fact, can we claim that the success rate of botulinum toxin A administration at the urethral level is statistically higher than non-administration? In the literature, for example, there are articles that argue there is no statistically significant difference between botulinum toxin administration and placebo, or that medical treatment with or without urethral botulinum toxin injection resulted in a reduction in Pdet and BOOI.

Reply: Thank you for the comment. The success rate of urethral BoNT-A injection for DV in women is not high in this study because we defined a VE improvement of > 20% and a GRA= 2 or 3 as a successful result. The results of this study provided objective evidence for a successful outcome, including improvement in Qmax, voided volume, PVR, VE, and BOOI. These improvements, however, are not observed in the failure group. (Table 4) We believe the treatment success is not a placebo effect. However, randomized control trial is still needed to clarify the fact.

In fact, the conclusions reached by the study are redundant with what is already present in the literature, except for the link between previous pelvic surgery and results of botulinum toxin A administration.

Reply: Thank you for the comment. We concluded the positive findings in this study.

Reviewer 3 Report

Comments and Suggestions for Authors

The work presented to me for review concerns a very important problem, which is non-neurogenic urination disorders, in the treatment of which botulinum toxin is used - a problem that is still current and very important. Non-neurogenic urination disorders are very difficult to treat and one of the methods is to inject the urethral sphincter with botulinum toxin.

Summary, materials and methodology written briefly and to the point, legibly. The results are presented reliably and very well graphically, especially videourodynamics. A very exhausting discussion. Literature from the last 20 years. It is difficult to find weaknesses in the article.

In my opinion, despite the unsatisfactory effectiveness of treatment of non-neurogenic urination disorder with botulinum toxin, the above work shows that at least half of the patients achieve positive results. It is an alternative treatment method to even less effective methods.

I propose that the article be accepted for publication without any comments.

Author Response

The work presented to me for review concerns a very important problem, which is non-neurogenic urination disorders, in the treatment of which botulinum toxin is used - a problem that is still current and very important. Non-neurogenic urination disorders are very difficult to treat and one of the methods is to inject the urethral sphincter with botulinum toxin.

Summary, materials and methodology written briefly and to the point, legibly. The results are presented reliably and very well graphically, especially videourodynamics. A very exhausting discussion. Literature from the last 20 years. It is difficult to find weaknesses in the article.

Reply: Thank you for the comment. Through more than 20 years, female DV remains a mystery in functional urology. We try to investigate the underlying pathophysiology and treatment strategy for this difficult to treat disease.

In my opinion, despite the unsatisfactory effectiveness of treatment of non-neurogenic urination disorder with botulinum toxin, the above work shows that at least half of the patients achieve positive results. It is an alternative treatment method to even less effective methods.

Reply: Thank you for the comment. Although the success rate of urethral BoNT-A injection on DV is not very high, this treatment still benefits more than 40% of women suffering from DV and difficulty in urination. These women had failed medical therapy or physiotherapy.

I propose that the article be accepted for publication without any comments.

Reply: Thank you for the comment.
